



# A New Inverse Modeling Approach for Emission Sources based on the DDM-3D and 3DVAR techniques: an application to air quality forecasts in the Beijing–Tianjin–Hebei Region

Xinghong Cheng[1], Zilong Hao[2], Zengliang Zang[2], Zhiquan Liu[3], Xiangde Xu[1], Yuelin Liu[4], Yiwen Hu[5], Xiaodan Ma[5]

1. State Key Lab of Severe Weather & Key Laboratory for Atmospheric Chemistry, Chinese Academy of Meteorological Sciences, Beijing 100081, China

2. Institute of Meteorology and Oceanography, National University of Defense Technology, Nanjing
211101, China

3. National Center for Atmospheric Research, Boulder, CO, USA

4. College of Architecture and Environment, Sichuan University, Chengdu 610065, China

5. Nanjing University of Information Science and Technology, Nanjing 210044, China

*Correspondence to*: Xinghong Cheng (cxingh@cma.gov.cn) and Zengliang Zang (zzlqxxy@163.com)

**Abstract.** We develop a new inversion method which is suitable for linear and nonlinear emission sources (ES) modeling, based on the three-dimensional decoupled direct (DDM-3D) sensitivity analysis module in the Community Multiscale Air Quality (CMAQ) model and the three-dimensional variational (3DVAR) data assimilation technique. We established the explicit observation operator matrix between the ES and receptor concentrations, and the background error covariance (BEC) matrix
of the ES which can reflect the impacts of uncertainties of the ES on assimilation. Then we constructed the inversion model of the ES by combining the sensitivity analysis with 3DVAR techniques. We performed the simulation experiment using the inversion model for a heavy haze case study in the Beijing-Tianjin-Hebei (BTH) region during December 27-30, 2016. Results show that the spatial distribution of sensitivities of $SO_2$ and $NO_X$ ES to their concentrations, as well as the BEC matrix of ES,
are reasonable. Using the *posteriori* inversed ES, underestimations of $SO_2$ and $NO_2$ during the heavy haze period are remarkably improved, especially for $NO_2$. Spatial distributions of $SO_2$ and $NO_2$ concentrations simulated by the constrained ES were more accurate compared with the *priori* ES in the BTH region. The temporal variations in regionally averaged $SO_2$, $NO_2$, and $O_3$ modelled concentrations using the *posteriori* inversed ES are consistent with in-situ observations at 45 stations over the BTH





region, and simulation errors decrease significantly. These results are of great significance for: studies

on the formation mechanism of heavy haze; reducing uncertainties of ES and its dynamic updating;

providing accurate 'virtual' emission inventories for air-quality forecasts and decision-making services

for optimization control of air pollution.

**1.   Introduction**

Since the implementation of the Air Pollution Prevention and Control Action Plan in September 2013,

urban air quality in China has improved overall. However, heavy haze frequently occurs over

Beijing-Tianjin-Hebei (BTH) and the surrounding region in winter. In recent years, many researchers

have studied the formation mechanism of heavy haze in the BTH region (Huang et al., 2014; Cheng et

al., 2016; Liu et al., 2016). These studies have shown that rapid conversion from primary gas pollutants

to particulates is an internal triggering factor for the "explosive" and "persistent" heavy haze (Wang et

al., 2014), and secondary particulate concentrations, such as sulfate and nitrate, account for a

significant percentage of $PM_{2.5}$. Thus, effectively controlling the emissions of precursors of secondary

aerosols (such as $SO_2$ and $NO_x$) is important for reducing environmental, economic, and human health

problems caused by $PM_{2.5}$ concentrations (Huang et al., 2014).

Emission inventories provide important fundamental data for investigating the causes of air

pollution, and atmospheric chemical transport model (ACTM). Uncertainties in ES are a major factor in

determining the simulated and forecast accuracy of the ACTM, and these uncertainties can greatly

affect the design of ES control strategies (Tang, 2006). The methods for establishing an emission

inventory include the bottom-up approach based on human activities, energy consumption statistics,

and various emission factors, as well as top-down inversion modeling of ES based on monitoring data

of air pollutants using satellite remote sensing and ground observations. Many studies have established

various ES inventories in China using the bottom-up approach, e.g., Bai et al., 1996; Streets et al., 2003;

Zhang et al., 2009, 2012; Cao et al., 2011; Zhao et al., 2012; Zhao et al., 2015; and Li et al., 2017.

However, the ES estimated by this method differ greatly due to large uncertainties in the statistical data,

emission factors, and spatiotemporal apportionment coefficients (Ma et al., 2004). Moreover, real-time

updates of emission inventories are difficult to achieve because of its rapid spatiotemporal variations

due to high-speed urbanization, and a delay in the release of statistical data of approximately 1–2 years.



The top-down approach is a useful supplement to bottom-up estimates, which are subject to

uncertainties in emissions factors and activities (Streets et al., 2003). Inverse modeling, in which

emissions are optimized to reduce the differences between simulated and observed data, is a powerful

method that eliminates the problems of the bottom-up approach.

Over the past decade, many researchers have tried to find an ideal inversion modeling tool that

improve the spatiotemporal distribution of ES. With the development of data assimilation technology

and ACTM, constraining the strength of ES using ACTM has become one of the main top-down

inversion methods (Enting, 2002; Sportisse, 2007). Researchers have primarily constrained the ES of

weak active chemical pollutants, such as $NO_x$, CO, $CO_2$, $SO_2$, $CH_4$, and CHOCHO using the following

methods: mass balance (Martin et al., 2003; Wang et al., 2007; Yang et al., 2011), back-trajectory

inverse modeling (Manning et al., 2011), adjoint modeling (Liu et al., 2005, Stavrakou et al., 2009;

Koohkan et al., 2013; Zhang et al., 2016; Zhai et al., 2018, and Wang et al., 2018), Bayes estimation

theory (Kopacz et al., 2009), ensemble Kalman-filtering (EnKF, Zhu et al., 2006, 2018; Barbu et al.,

2009; Tang et al., 2011, 2016; Miyazaki et al., 2012; Wang et al., 2016; Peng et al., 2017), the

four-dimension variational (4DVAR) technique (Elbern et al., 2000, 2007; Gilliland et al., 2006;

Napelenok et al., 2008; Henze et al., 2008; Corazza et al., 2011; Jiang et al., 2011), an adaptive nudging

scheme in the CMAQ model (Xu et al., 2008; Cheng et al., 2010), inversion algorithms combining

pollutant dispersion models and a Monte Carlo simulation (Yang et al., 2013), and sensitivity analysis

(Fu et al., 2007; Hu et al., 2009; Mijling et al., 2012). Results show that using an inversion modeling

approach to retrieve the spatial distribution of ES can greatly improve air quality simulations and

forecasts by the ACTM. Many studies have achieved a certain amount of improvement using the EnKF

and 4DVAR methods. The advantage of the EnKF method is that the observation operator is implicit in

the assimilation process of ES, and it avoids developing the tangent linear and adjoint models.

However, this method has stricter requirements for error perturbations in ES and the construction of

bias-correction models. In addition, the large number of ensemble members in the EnKF method leads

to a huge computational cost. Some studies adopted the 4DVAR method to inverse the ES of $NO_x$ and

CO based on the Goddard Earth Observing System (GEOS)-Chem adjoint model. However, the

GEOS-Chem model is often used to simulate large-scale physical and chemical processes and rarely



utilized in urban air quality forecasts. This method also has high computational costs due to the gradient calculation of the objective function. In addition, the EnKF and 4DVAR methods exhibit difficulty in accurate inversion of ES in real applications due to the absence of sensitivity analysis of

the source-receptor (S-R) relationship.

The 3DVAR method is a generalization of optimal interpolation methods. It has the advantages of conveniently adding dynamical constraints and directly assimilating unconventional observation data (Li et al., 2013). 3DVAR is widely-used in the assimilation of meteorological and atmospheric chemical data due to simplicity, ability to use complex observations operators, and low computational cost.

However, this method has two requirements: assimilated variables must remain relatively stationary within the assimilation window, and the method must be coordinated between the assimilated initial field and the iterative integration of the model. To apply the 3DVAR method to inverse ES, it is necessary to construct an inversion model that can satisfy the aforementioned requirements. Firstly, although ES have monthly, seasonal, and annual variations, the variation of ES is constant within a

short period (e.g., for an assimilation window of 1 hour). Secondly, the assimilation effect of ES depends on the quality of observation data and the consistency between observed and simulated values. To ensure consistency between observations and simulations, the sensitivity of the receptor's concentrations with respect to the ES should be accurately calculated (Hu et al., 2009). Using the three-dimensional decoupled direct (DDM-3D) sensitivity analysis method within the CMAQ model,

reasonable sensitivity coefficients between ES and the receptor's concentrations can be calculated. This coefficient matrix is then used in the 3DVAR assimilation process, which ensures consistency between the ES and the modelled results. Thus, the top-down 3DVAR constraint methods for ES based on the first- or high-order DDM-3D sensitivity analysis techniques can maintain the coordination between assimilated field of ES and simulated concentration of air pollutants.

The primary methods used to calculate the S-R relationship include the brute force, the adjoint, and the DDM-3D method. Many studies have shown that these methods can improve ES inventories constructed by bottom-up methods for $NO_x$ (Napelenok et al., 2008), CO (Bergamaschi et al., 2000 and Heald et al., 2004), NH3 (Gilliland et al., 2003), and EC (Hu et al., 2009). The adjoint method is a backward-sensitivity calculation method, while brute force and DDM-3D are forward-sensitivity


calculation methods. For inverse modeling of pollution sources with single receptor, the

backward-sensitivity method is more suitable, with low computation costs for certain grid sizes in a

given time period, but it is not suitable for ES with multiple receptors, which result in high

computational costs. The forward-sensitivity calculation method is more suitable for inversing the ES

based on observed data from satellites or multiple surface stations (Hu et al., 2009). Cohan et al. (2002)

introduced the DDM-3D method to the CMAQ model, and created the CMAQ-DDM-3D module for

low-order sensitivity calculations in early 2010. In 2014, they added a high-order calculation module for

particles (High-Order DDM-3D for Particular Matter; HDDM3D/PM) in the newly released version of

CMAQ model. Wang et al. (2013) claim that the sensitivity calculation results using the DDM-3D

method are more reasonable than the brute force method. Some studies have used the DDM-3D method

(Napelenok et al., 2008; Hu et al., 2009) or a combination of the DDM-3D and a discrete Kalman-filter

method (Wang et al., 2013) in conjunction with measurements from satellite and ground observations to

inverse BC and $NO_x$ ES in the United States. Because inverse modeling of ES based on discrete

Kalman-filtering is more suitable for linear systems, we use the DDM-3D method to calculate the S-R

linear and non-linear relationship.

130          The results of inverse modeling are very sensitive to uncertainties in the ES of NOx, $NH_4$ and

inorganic aerosols (Zhang et al., 2016). Impact of uncertainties in the ES on the assimilation effects

need to be considered in the top-down inversion model. The top-down 3DVAR inversion methods

developed in this study can include the impacts of ES uncertainties by the BEC matrix of ES based on

multiple sets of ES. We developed a new inverse modeling approach for ES that combines the

DDM-3D sensitivity analysis method with the 3DVAR assimilation technique, and then applied it to a

case study during a typical heavy haze episode. This paper is organized as follows: Section 2 describes

the inversion model and presents results of sensitivity analysis and the BEC; Section 3 provides details

of the WRF-CMAQ model, and configurations and experiments of simulation; Section 4 presents the

results of the control and experiment simulations with the *priori* and constrained *posteriori* ES,

respectively; finally, the discussions and conclusions are provided in Section 5.

## 2. Model and data

We used an offline modeling system that includes two components: the Weather Research and



Forecasting (WRF) model (Michalakes et al., 2004) and the CMAQ model (Dennis et al., 1996; hereafter

referred to as WRF-CMAQ). This study focuses on the BTH region with 5 x 5-km grid spacing, 32

vertical layers of varying thickness (between the surface and 50 hPa), and an output interval of 1 h. The

WRF-CMAQ simulations are driven by the National Center for Environmental Prediction Final (NCEP

FNL) analysis data every 6 h during December 27-30, 2016 and the Multi-resolution Emission Inventory

for China (MEIC) data for 2012, with 1 °×1 ° and 0.25 °×0.25 ° grid spacing, respectively. The CMAQ

model was configured to utilize all layers from the input meteorology. Emissions datasets for CMAQ

were generated by the Sparse Matrix Operator Kernel Emissions (SMOKE) model developed by the

University of North Carolina (UNC, 2014). Meteorological outputs from the WRF simulations were

processed to create model-ready input to CMAQ using the Meteorology-Chemistry Interface Processor

(MCIP; Otte al., 2010). The boundary conditions for chemical trace gases consisted of idealized,

northern hemispheric, mid-latitude profiles based upon output from the National Oceanic Atmospheric

Administration (NOAA) Agronomy Lab Regional Oxidant model. The model simulation started on

December 27, 2016. To assess the improved effects of inverse modeling of ES during the heavy haze

episode in December 2016, we ran two simulations: a control run with the *priori* MEIC data for 2012,

and an experiment run with *a posteriori* constrained ES.

Hourly measurements of $SO_2$, $NO_2$, and $O_3$ concentrations at 129 stations during December 27–30,

2016 were obtained from the China National Environmental Monitoring Centre. These data are used to

validate simulations from the control and experiment runs. The simulation domains and the locations of

the 129 stations are shown in Figure 1.

**3.  Inverse modeling method**

**3.1 Constructing BEC matrix**

To construct the BEC matrix for the inversion model, we combined the National Meteorological Center's

(NMC) technique (Parrish and Derber, 1992) with the SMOKE model based on uncertainty analysis of

the ES inventories. We created the BEC matrix by four steps, as follows:

(1)Determine the total errors of ES from a priori bottom-up inventory.

Uncertainty analyses of ES require detailed information of activities and emission factors from the


priori MEIC emission inventory. The relevant data collected in the China Environmental Yearbook are limited, and do not satisfy with the requirements of uncertainty analysis. Therefore, we used the available research results relating to $SO_2$ and NOx ES, and conducted uncertainty analysis for four types of major

sources (industry, power plants, residents, and transportation), and determining the error ranges in total emission rates of $SO_2$ and NOx (Table 1). Uncertainties in $SO_2$ industry and power plant ES are slightly greater than those for NOx, while the opposite is true for emissions from the residential and traffic sectors.

(2)Generate multiple sets of inventories using the random perturbation technique.

Based on the aforementioned error ranges in total emission rates, we generated 30 sets of inventories for $SO_2$ and NOx with the same resolution as MEIC for each month using a random perturbation method.

(3) Process the 3-D gridded ES as input to the CMAQ model.

We used the SMOKE model, national population and road network distribution data in 2016, the temporal apportionment coefficients in the BTH region (Zhang et al., 2007, 2009, Simpson et al., 2003,

and Wang et al., 2010), and the CB05-ae06-aq chemical species data in the CMAQ model, to process thirty sets of nationwide emission inventories into 3-D gridded ES with a grid spacing of 5×5 km. Each grid has 124×130 points, with 12 vertical levels.

(4) Calculate the BEC matrix of each 3-D gridded ES.

Finally, the NMC method was used to calculate the BEC matrix of the 3-D gridded ES for each month,

including horizontal and vertical correlation coefficients and standard deviations. The background error is defined as the difference between thirty sets of 3-D gridded ES generated by the random perturbation method, and the 3-D gridded background ES directly processed from the original MEIC emission inventory with the SMOKE model, at every hour (24-h strengths of ES for each month).

According to the literature (Liu et al., 2011; Li et al., 2013; Zang et al., 2016), the approximate calculation of the BEC matrix is as follows:

$$\mathbf{B} \approx \frac{1}{2} \langle (e_t - e_b)(e_t - e_b)^T \rangle, \tag{1}$$

where $e_t$ is the perturbation field and $e_b$ is the background field of *a priori* ES. Eq. (1) can be written as follows:

$$\mathbf{B} = \mathbf{DCD}^T, \qquad\qquad\qquad (2)$$

where $\mathbf{D}$ is the standard deviation (SD) matrix and $\mathbf{C}$ is the correlation coefficient matrix. With this

factorization, $\mathbf{D}$ and $\mathbf{C}$ can be calculated separately. $\mathbf{D}$ is a diagonal matrix whose elements are SD of

all state variables in the 3-D grids. $\mathbf{D}$ is used to improve the ability of the 3DVAR in representing the

impacts of local emissions at one grid on other grids; these impacts vary in the vertical direction, and

they are heterogeneous in the horizontal direction.

Figure 2 shows the spatial distribution of averaged emission rates for thirty sets of 3-D gridded ES,

and the SD of the BEC matrix for $SO_2$ and $NO_x$ ES at 08:00 local time in December. $SO_2$ and $NO_x$ ES

have different spatial distributions in terms of average strength and standard deviation. The $NO_x$

emissions are mainly concentrated in cities and surrounding areas, and they are much greater in Beijing,

Tianjin, and Shijiazhuang than other cities. The $SO_2$ emissions are mainly concentrated in Shijiazhuang,

Jinan, the north and east of Shanxi Province and their surrounding areas. Figure 3 shows variations in

the horizontal correlation coefficients by grid distance, and the vertical distributions of the SD in $\mathbf{B}$ for

$SO_2$ and $NO_x$ ES in December 2012. The cross between the correlation curve and the $e^{-1/2}$ line

(dashed line) represents the horizontal length scale ($L_s$), and the $L_s$ of the two species falls between five

and six grid distances. Namely, the horizontal scale felt is approximately 25–30 km. The correlation

coefficient of $SO_2$ is slightly larger than that of $NO_2$. The difference in the correlation coefficients

between $SO_2$ and $NO_x$ ES increases with grid distance and this is related to the regional pollution

characteristics of $SO_2$. The vertical distributions of the SDs in $\mathbf{B}$ for $SO_2$ and $NO_x$ ES vary with height:

the SD of $SO_2$ ES are larger on the fourth and eighth model levels than on other levels; while for $NO_x$

ES, the SD on the first level is the largest, that on the eighth level take the second place, and the SDs on

all other levels are smaller.

### 3.2 Sensitivity analysis

The sensitivity analysis module (DDM-3D) in CMAQ solves a series of equations while simultaneously

calculating pollutant concentrations. The local sensitivity of pollutant concentrations with respect to

several specified parameters, such as ES, initial and boundary conditions, and chemical reaction rates,

can be calculated by the DDM-3D method. The sensitivity equations about the ES are solved using the

governing equations of the model, as follows (Hu et al., 2009):


$$S_j = P_j \frac{\partial C}{\partial P_j} = P_j \frac{\partial C}{\partial(\varepsilon_j P_j)} = \frac{\partial C}{\partial \varepsilon_j} \qquad , \tag{3}$$

where $S_j$ is the sensitivity of the pollutant j to the parameter $P_j$, $P_j$ is the *priori* ES of the pollutant *j*, *C* is

the concentration of the pollutant *j*, and $\varepsilon_j$ is the perturbation coefficient of the ES. Theoretically, the

DDM-3D method truly captures the sensitivities of pollutant concentrations to ES, and results are more

accurate than the brute force method, for the BTH region (Wang et al, 2013). In addition, the results of

the DDM-3D method are more accurate and efficient for highly nonlinear pollutants (such as $O_3$ and

$PM_{2.5}$) and small perturbations.

We used the WRFv3.7.1 and CMAQv5.0.2-DDM-3D models as well as 3-D gridded *a priori* ES

from MEIC in 2012 to calculate the sensitivity coefficients of $SO_2$ and $NO_2$ concentrations with respect

to ES during the "heavy haze" episode of December 27–30, 2016. Figure 4 shows the spatial

distribution of 96-h averaged sensitivity coefficients for $SO_2$ and $NO_2$ concentrations with respect to ES

during December 27–30, 2016. The sensitivity coefficients of $SO_2$ and $NO_2$ concentrations all exhibit

inhomogeneous distribution. The sensitivity coefficients are higher in Beijing, Shijiazhuang, Baoding,

and surrounding regions, i.e., $SO_2$ and $NO_2$ concentrations in those areas are greatly affected by the $SO_2$

and $NO_x$ ES.

### 3.3 Observation operators

The relationship between pollutant source and the receptor's concentration is established according to

Eq. (3). Next, we create the observation operator matrix between ES and receptor concentrations as

follows:

$$\mathbf{H} = \frac{\partial \mathbf{c}}{\partial E} = \frac{\partial \mathbf{c}}{\partial(\varepsilon_j E_0)} = \frac{S_j}{E_0}, \tag{4}$$

where $\mathbf{H}$ is the observation operator matrix, $E$ are the *posteriori* ES, which can be written as the product

of the perturbation coefficient and *a priori* ES during the assimilation window time, and $E_0$ are the

*priori* ES. For primary pollutants such as $SO_2$ and $NO_2$, $S_j$ is a first-order sensitivity coefficient, and $\mathbf{H}$

is a linear observation operator between the ES and the receptor concentration. For secondary pollutants

such as $PM_{2.5}$ and $O_3$, $S_j$ is a high-order sensitivity coefficient, and $\mathbf{H}$ is a nonlinear observation operator.

In this study, we use the first-order sensitivity coefficient to calculate $\mathbf{H}$ for $SO_2$ and $NO_x$ ES.



### 3.4 Observational error covariance

We firstly performed quality control on the observed $SO_2$ and $NO_2$ concentration data. This process involved three steps:

    (1) Redundant data removal, and matching the density of observation data to the model grid. For some grids with more than one observation station, we used the average of those stations.

    (2) Extrema control, i.e., filtering out data exceeding three times of SD of observation data.

260        (3) Anomaly removal, i.e., data that remained constant for 24 consecutive hours, as well as any negative data, were removed.

Data that passed quality control still contained observation or instrument errors. These errors are related to many factors such as instrument type, calibration design, and environmental conditions. In
addition, in the variational assimilation process, representation errors caused by the forward-calculation and variational processes must be considered. Higher-resolution models produce smaller representation errors. Representation error, $\varepsilon_r$, can be expressed as follows (Pagowski et al., 2010):

$$\varepsilon_r = \gamma \varepsilon_o \sqrt{\frac{\Delta x}{L_s}}$$
(5)

where $\gamma$ is the amplification factor, which is used to adjust the instrument error, $\varepsilon_o$ is related to the
$SO_2$ and $NO_2$ concentrations, and $\Delta x$ is grid distance of the model. Note that $L_s$ is usually smaller in urban areas and larger in suburban areas. The amplification parameters of the observing stations in cities, suburbs, and rural areas are 2.5 km, 5 km, and 10 km, respectively (Zang et al., 2016). Finally, the total observation error for the $SO_2$ and $NO_2$ concentrations, $\varepsilon$, is written as:

$$\varepsilon = \sqrt{\varepsilon_o^2 + \varepsilon_r^2}.$$
(6)

### 3.5 3DVAR inversion model

We introduce a cost function with respect to the ES in accordance with 3DVAR:

$$J(e) = \frac{1}{2}(e - e_b)^T \mathbf{B}^{-1}(e - e_b) + \frac{1}{2}(\mathbf{H}e - c)^T \mathbf{R}^{-1}(\mathbf{H}e - c),$$
(7)

where $c$ is the observation variable, $\mathbf{R}$ is the observation error matrix, and $e$ is the inversing variable of an *a posteriori* ES. The optimal inversion of ES for $SO_2$ and $NO_x$ are obtained using Eq. (7). The
3DVAR solves for the minimum value of $J(e)$ to determine the inversing variable $e$. This process typically employs a gradient propagation method, with the increment of an ES defined as follows:

$$\delta e = e - e_b,$$
(8)

Accordingly, the innovation vector of pollutant concentration is defined as:





$$\delta c = c - \mathbf{H}e_b. \tag{9}$$

Therefore, Eq. (7) can be written in gradient form:

$$J(\delta e) = \frac{1}{2}\delta e^T \mathbf{B}^{-1}\delta e + \frac{1}{2}(\mathbf{H}\delta e - \delta c)^T \mathbf{R}^{-1}(\mathbf{H}\delta e - \delta c). \tag{10}$$

After conditionally processing the cost function, a finite-memory quasi-Newton method was used to conduct iterative minimization. The background field was set as the initial iteration values. The maximum number of steps at the end of the iteration and the minimum gradient for convergence were

predetermined. The iteration was finished when one of these conditions was met, and the optimal analysis increment, $\delta e$, was obtained. Finally, the optimal assimilation analysis field of the ES, $e = \delta e + e_b$, was obtained. The result was a three-dimensional variational inversion model of the ES, using the uncertainty analysie of the ES and sensitivity coefficients between the ES and receptor's concentrations; the overall framework is shown in Figure 5.


## 4.    Results and discussion

A typical heavy haze event occurred in the BTH region at the end of December 2016. We applied the 3DVAR inversion model to constrain the hourly *posteriori* ES of $SO_2$ and $NO_2$ using measurements from 45 and 129 stations, respectively, on December 27, 2016. We validated simulations from the

control and experiment run using observational data during December 28–30, 2016.

Figure 6 shows time series of hourly, regional averaged $SO_2$ and $NO_2$ simulations from the control run, observations, and sensitivity coefficients at 45 stations in the BTH region during December 27–30, 2016. The trends in modelled concentrations and sensitivity coefficients of $SO_2$ and $NO_2$ concentrations with respect to ES are consistent, therefore the sensitivity coefficients can reasonably

reflect the impacts of the ES on concentrations. However, simulated $SO_2$ and $NO_2$ concentrations with the underestimated *priori* ES are all significantly lower than observations during the heavy haze period. Thus, it is important to improve the *priori* ES using the inversion model.

Figures 7 and 8 show the spatial distributions of 24-h averaged emission rates from the *priori* and *posteriori* ES of $SO_2$ and $NO_2$, and their increments on December 27, 2016. Emission rates of the $SO_2$

and $NO_2$ *posteriori* ES in the major cities and surrounding areas clearly increase. Compared with the *priori* ES, the maximum strengths of $SO_2$ and $NO_2$ ES increase by approximately 17% and 500%,


respectively. Therefore, the strengths of SO$_2$ and NO$_2$ in the *priori* ES were greatly underestimated, especially for NO$_2$.

Using the WRF-CMAQ and *posteriori* ES, we simulated concentrations of SO$_2$, NO$_2$, and O$_3$ in the BTH region during December 28–30, 2016, and validated these simulations with measurements from 45 stations. Figures 9 and 10 show the spatial distributions of 72-h averaged SO$_2$ and NO$_2$ concentrations simulated with the *priori* and *posteriori* ES, increments, and their observations. In general, SO$_2$ and NO$_2$ concentrations simulated using the *posteriori* ES are closer to observations than the *priori* ES, and regional differences in improvements for SO$_2$ and NO$_2$ exist. For SO$_2$, the improvement is noticeable in the BTH region. However, the simulated concentrations in Beijing with the *posteriori* ES are overestimated. This may be related to greater uncertainties in SO$_2$ sources and the impacts of regional transport from surrounding areas. For NO$_2$, simulated differences with the *priori* and *posteriori* ES are significant in major cities such as Beijing, Tianjin, Shijiazhuang, Baoding, Xingtai, Handan, and Jinan. The simulated concentrations of NO$_2$ using the *posteriori* ES are more consistent with measurements, while those with the *priori* ES are significantly underestimated.

We also investigated temporal variations in regionally-averaged SO$_2$, NO$_2$, and O$_3$ concentrations simulated using the *priori* and *a posteriori* ES, and observations from the 45 stations over the BTH region during December 28–30, 2016 (Figure 11). In general, simulated SO$_2$, NO$_2$, and O$_3$ concentrations using the *posteriori* ES are closer to measurements, while the SO$_2$ and NO$_2$ concentrations simulated by the *priori* ES are significantly lower than observations, and the modelled O$_3$ concentrations are obviously higher than measurements. In addition, the peak of SO$_2$ simulations with the *posteriori* ES are close to measurements, but the peak of NO$_2$ and valley of O$_3$ simulations are lower and higher than observations, respectively. This may be related to the absence of inverse modeling of volatile organic compound (VOC) ES and uncertainties of sensitivity coefficients calculation. In this study, we used only first-order sensitivity coefficients, but the relationship between ES of precursors of O$_3$ such as VOCs and NOx, and their receptor's concentrations are nonlinear, and O$_3$ is generated from both NOx and VOCs ES. Therefore, higher-order sensitivity coefficients are necessary for inverse modeling of ES of NOx and VOCs.

To further assess the simulated accuracy of SO$_2$, NO$_2$, and O$_3$ concentrations, we calculated the





following statistics (Willmott et al., 2011): correlation coefficient (R), root-mean-squared error (RMSE),

mean bias (MB), normalized mean bias (NMB), and index of agreement (IOA; see Table 2). Except that

R of $NO_2$ and $O_3$ decrease and RMSE of $O_3$ increases using the constrained ES, other statistics show

improvements. Especially, MB and NMB of three pollutants decline significantly and IOA are closer to

1.0, which means that modelled results of three pollutants are more consistent with observations. R

between $SO_2$ simulation and observation shows a slight improvement when using the *posteriori* ES,

whereas R decreases for $NO_2$ and $O_3$, and it may be related with the absence of constraint of VOCs ES.

### 5.    Summary and conclusions

We developed a new inverse approach of ES by combining the sensitivity analysis technique between

ES and receptor's concentration, and the 3DVAR method. Our approach is suitable for solving the

linear or nonlinear inversion problems for ES, and it compute fastly and obtain the relatively accurate

real-time dynamic updates of ES. First, we used the sensitivity analysis tool in the CMAQ model to

construct the explicit observation operator matrix between ES and receptor's concentration. Next, we

created the BEC matrix for ES based on uncertainty analysis and the NMC statistical method. Finally,

we established a three-dimensional variational inverse method of ES based on the observation operator

and BEC matrix.

     The 3DVAR inversion model was applied to a heavy haze case study in the BTH region during

December 27–30, 2016. Results show that the observation operators between $SO_2$ and $NO_2$ ES, and

their concentrations, as well as spatial distributions of the BEC matrix are both reasonable. Using the

3DVAR inversion model, the *priori* $SO_2$ and $NO_2$ ES improved obviously during the heavy haze

process, especially for $NO_2$ ES. The spatial distributions of $SO_2$ and $NO_2$ concentrations simulated

using the *posteriori* ES are more consistent with measurements than the *priori* ES, especially in major

cities over the BTH region. Simulation errors of $SO_2$, $NO_2$ and $O_3$ concentrations with the *posteriori* ES

significantly decrease, whereas simulations of three pollutants using the *priori* ES are underestimated.

Future studies will include the applicability and accuracy of this method for different seasons and

regions, and different chemical species such as other primary pollutants (e.g., CO) and precursors of

secondary pollutants (e.g., $PM_{2.5}$, $PM_{10}$ and $O_3$). An emphasis may be placed on constructing the

nonlinear explicit observation operator for precursors of secondary pollutants such as VOCs ES using



the high-order sensitivity analysis technique, and assessing improvement effects of the *posteriori* ES

with the 3DVAR inversion method and CMAQ model.

*Data availability.* The NCEP-FNL reanalysis data are publicly available at http://rda.ucar.edu/datasets/ds083.2/. The $SO_2$, $NO_2$, and $O_3$ measurements are available at http://113.108.142.147:20035/emcpublish.


*Author contributions.* XC and ZZ designed the research. XC and ZH constructed the 3DVAR inversion model, designed model experiments and performed simulations. XC, ZH, ZZ, YL, YH and XM contributed to the data processing and analyses. XC and ZH analyzed the results and wrote the paper with inputs from all authors. ZL and XX contributed to theoretical direction for establishing the inversion

model.

*Competing interests.* The authors declare that they have no conflicts of interest.

*Acknowledgements.* We are grateful to Tsinghua University for providing the emission inventory and the

China National Environmental Monitoring Centre for providing surface $SO_2$, $NO_2$, and $O_3$ observation data.

*Financial support.* This work was supported jointly by the Fundamental Research Funds for Central Public-interest Scientific Institution from Chinese Academy of Meteorological Sciences (grant no.

2016Y005), the National Natural Science Foundation of China (grant no. 91644223), and the National Research Program for Key Issues in Air Pollution Control (grant no. DQGG0104).

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




Table 1. Uncertainty of $NO_x$ and $SO_2$ values used in the SMOKE model and calculation of the BEC matrix.

| Categories | $NO_x$ | $SO_2$ |
|---|---|---|
| Industry | (−32.4%, 33.0%) | (−37.5%, 38.8%) |
| Power | (−32.4%, 33.0%) | (−37.5%, 38.8%) |
| Residential | (−30.0%, 34.0%) | (−15.0%, 16.0%) |
| Transportation | (−55.4%, 70.3%) | (−17.0%, 20.0%) |

Table 2. Statistics for simulated $SO_2$, $NO_2$, and $O_3$ from control and experiment runs using the *priori* and *posteriori* inversed ES at 45 stations in the BTH region during December 28–30, 2016. Bold type indicates better statistical results.

| Parameters | Control Run | | | Experiment Run | | |
|---|---|---|---|---|---|---|
| | $SO_2$ | $NO_2$ | $O_3$ | $SO_2$ | $NO_2$ | $O_3$ |
| R | 0.80 | 0.82 | 0.89 | **0.82** | 0.52 | 0.87 |
| RMSE | 14.61 | 8.89 | 5.02 | **6.60** | **8.68** | 6.31 |
| MB | -40.98 | -48.20 | 26.91 | **3.23** | **-2.23** | **4.70** |
| NMB | -0.61 | -0.81 | 1.78 | **0.05** | **-0.04** | **0.31** |
| IOA | 0.42 | 0.27 | 0.45 | **0.89** | **0.68** | **0.84** |










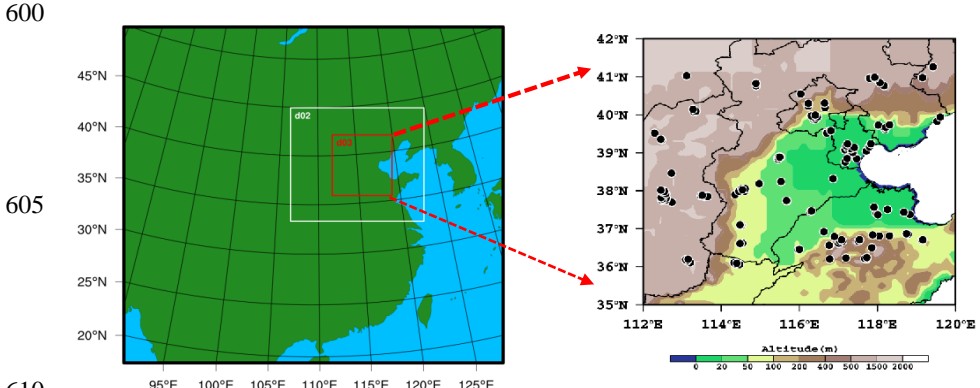



**Fig. 1** (a) Domain of the WRF-CMAQ model and (b) location of environmental monitoring stations in the innermost domain over the BTH region.

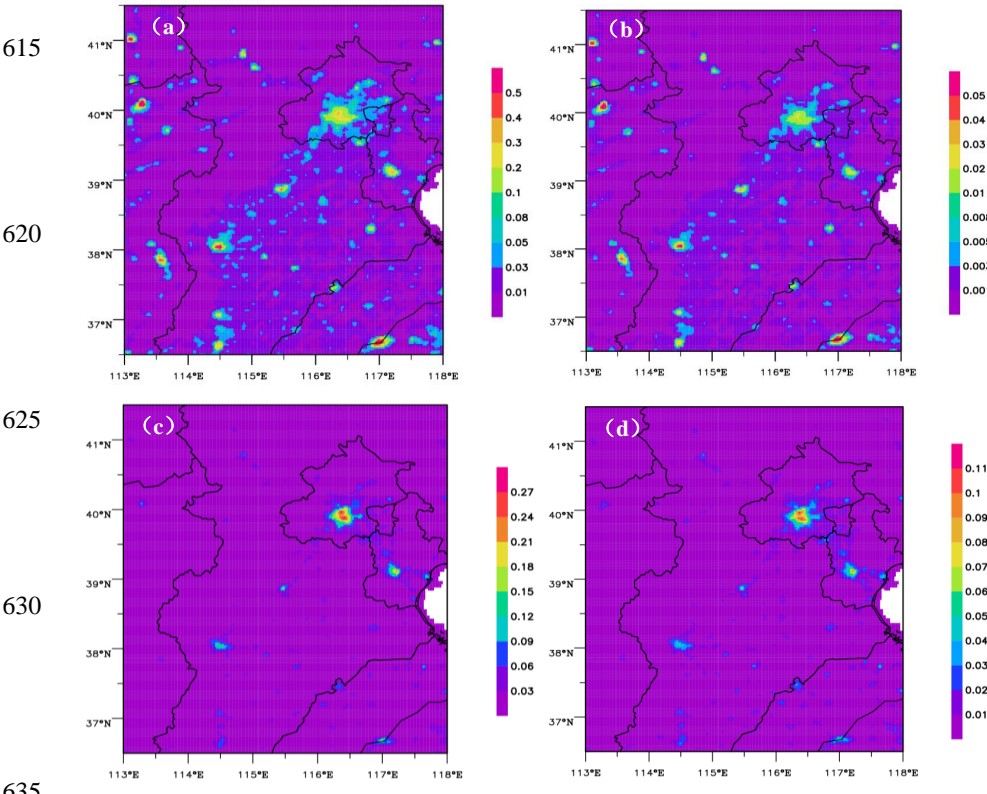






**Fig. 2** Spatial distributions of (a) averaged emission rates of $SO_2$ ES, (b) standard deviation in the BEC of $SO_2$ ES, (c) averaged emission rates of NOx ES, (d) standard deviation in the BEC of NOx ES at 08:00 local time in December 2012. Unit: mole/s.

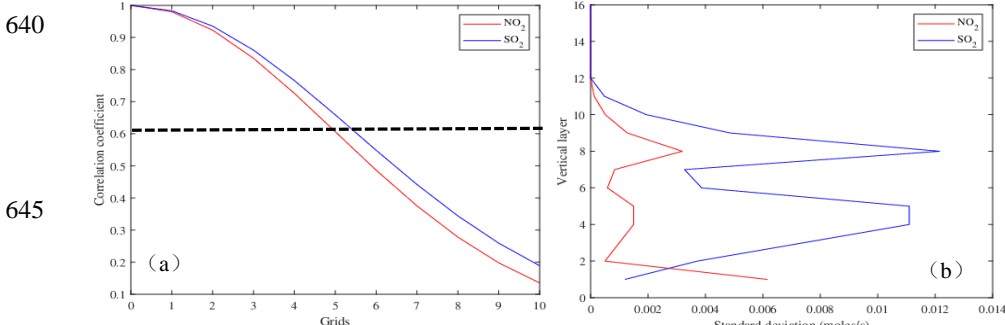

**Fig. 3** (a) Horizontal correlation coefficients with increasing grid distance, and (b) vertical profiles of standard deviations in the BEC of $SO_2$ and $NO_2$ ES in December 2012. Dashed line is the baseline of horizontal correlation scale.

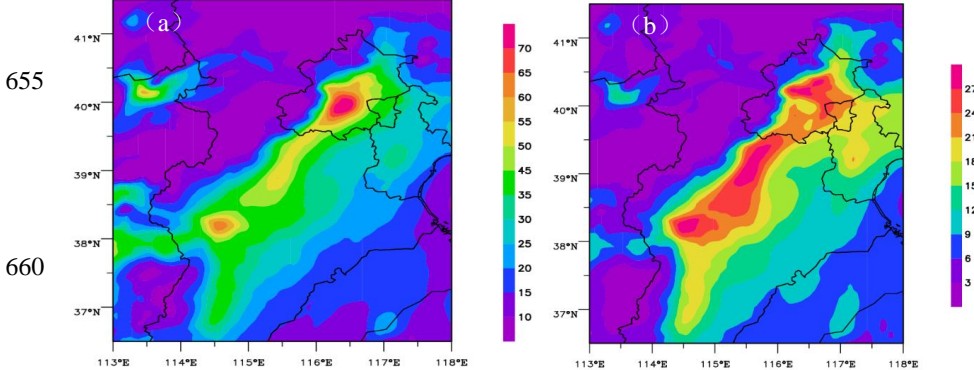

**Fig. 4** Spatial distributions of 96-h averaged sensitivity coefficients ($\mu g\ m^{-3}$) of (a) $SO_2$ and (b) $NO_x$ concentrations with respect to $SO_2$ and NOx ES during December 27–30, 2016.

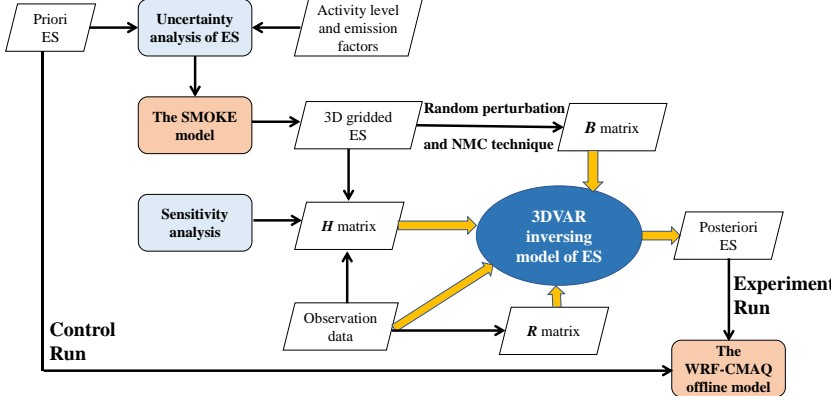

**Fig. 5** Flowchart of the 3DVAR inversion model of ES and simulation experiments.

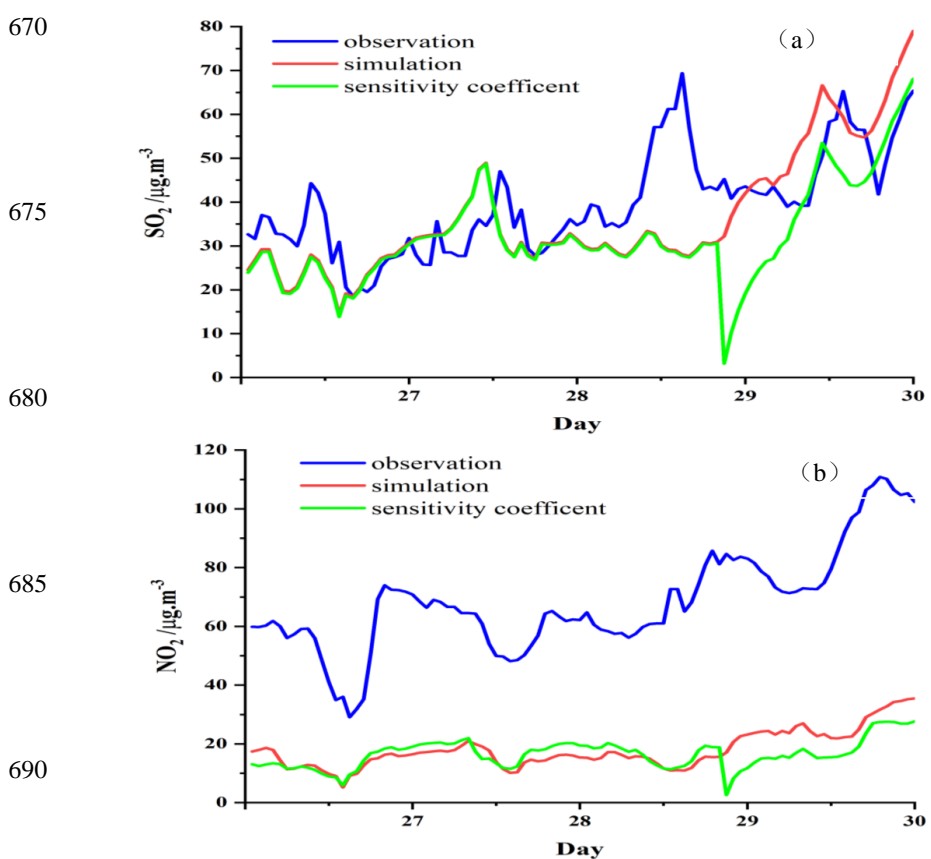

**Fig. 6** Time series of hourly, regionally-averaged (a) SO₂ and (b) NO₂ simulations with the *priori* ES, observations, and the first-order sensitivity coefficients between the ES and receptor's concentration at 45 stations over the BTH region during December 27-30, 2016.

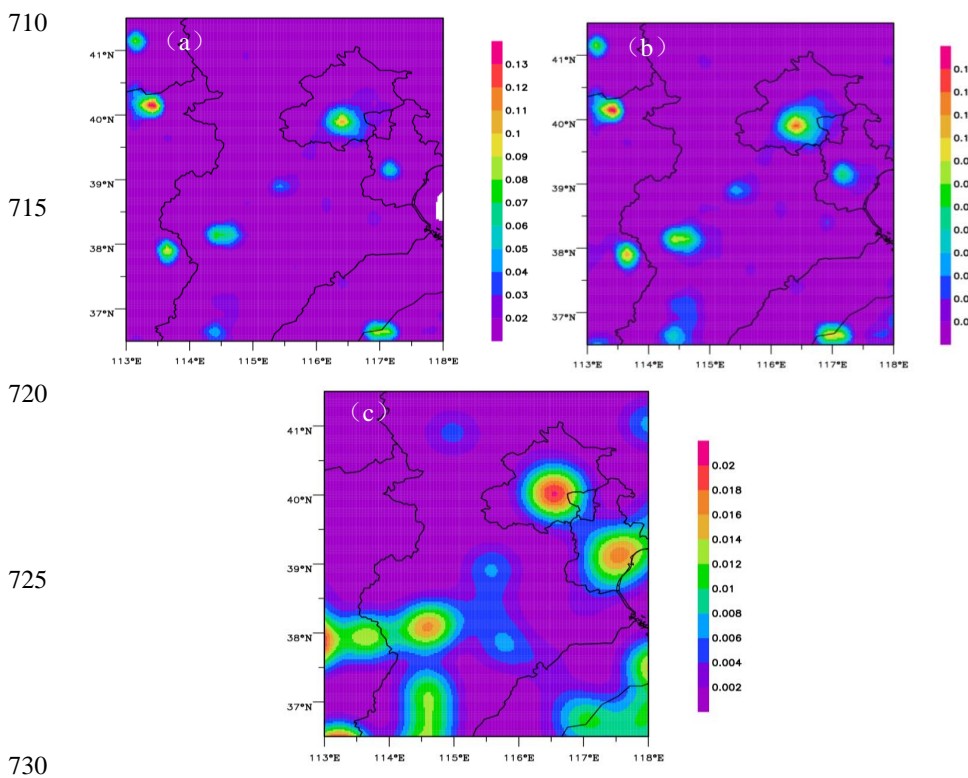

**Fig. 7** Spatial distributions of 24-h averaged emission rates for SO$_2$ (mole/s) from the (a) *priori* and (b) *posteriori* ES, and (c) the increment on December 27, 2016.



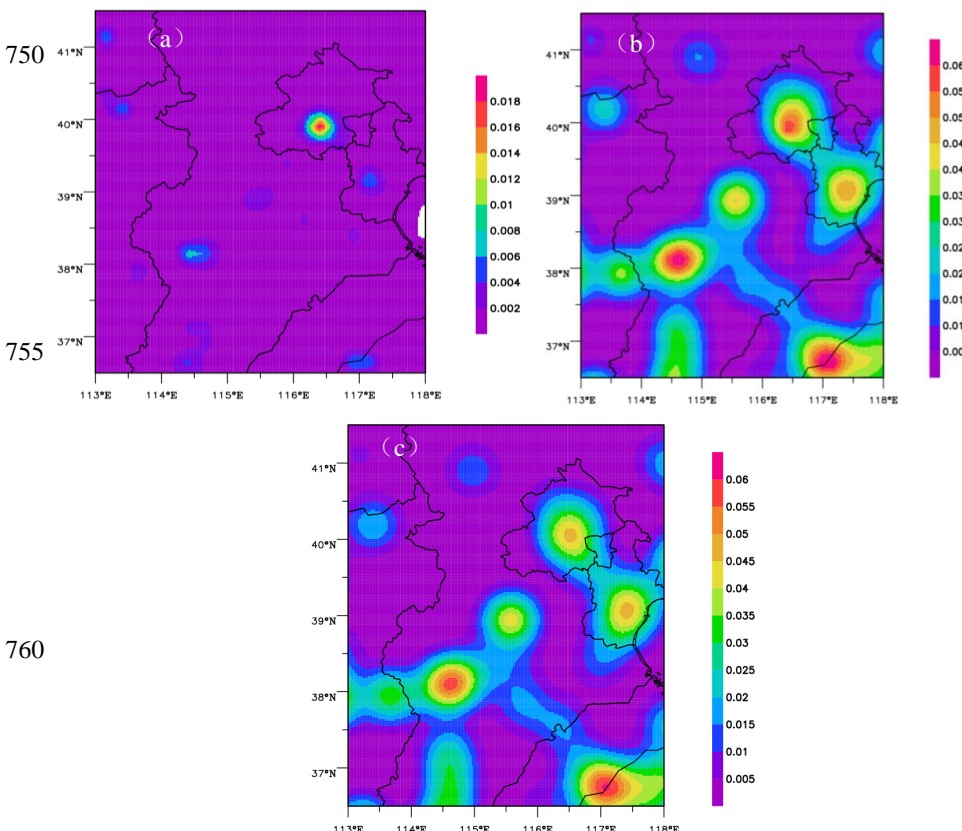

**Fig. 8** Same to Fig.7 except for NO₂.



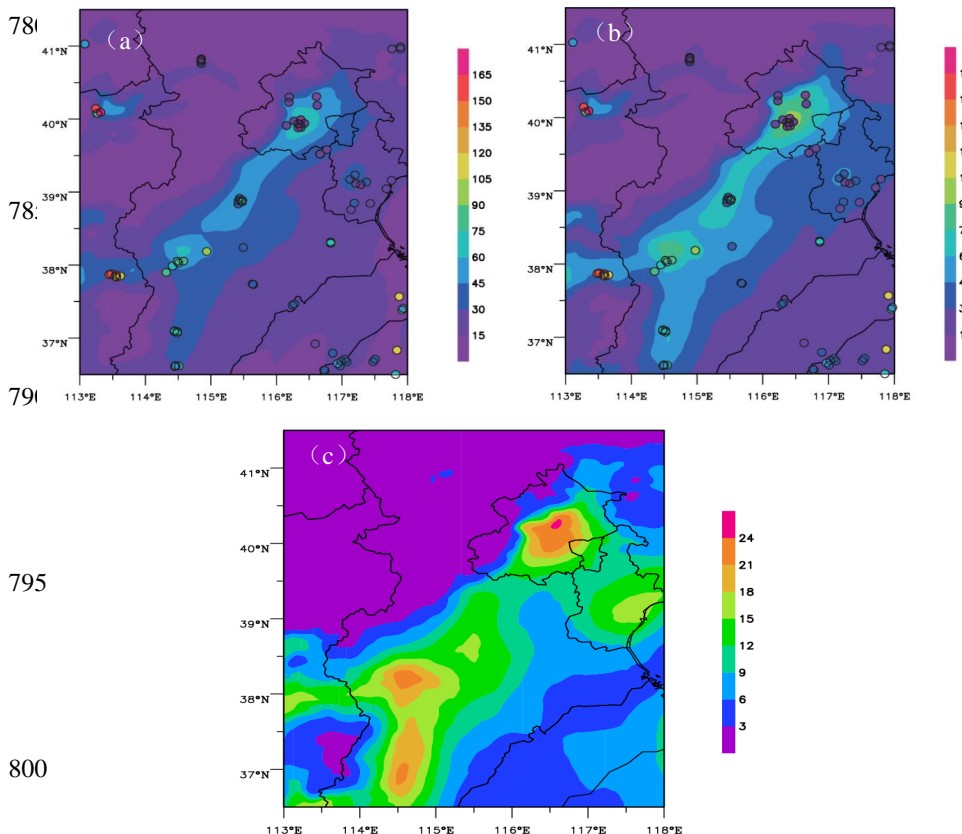

**Fig. 9** Spatial distribution of 72-h averaged SO$_2$ concentration simulated with the (a) *priori* and (b) *posteriori* ES, and (c) the increment during December 28-30, 2016. Color solid dot denotes the measurements.





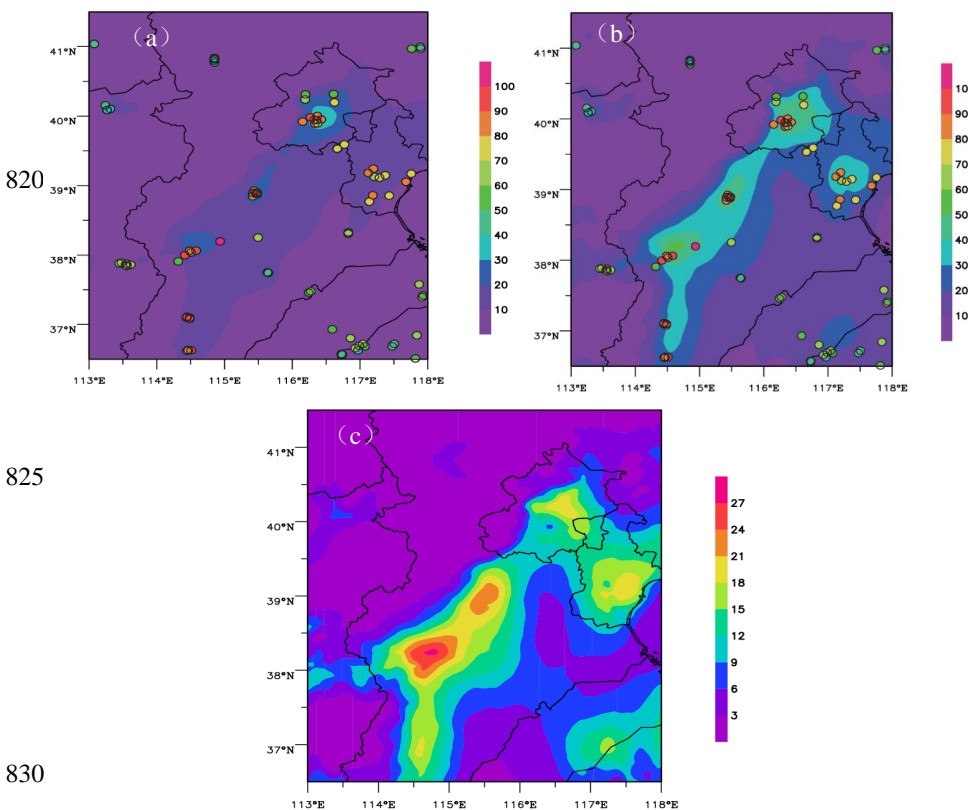

**Fig. 10** Same to Fig.9 except for NO₂.

850

855

860

865

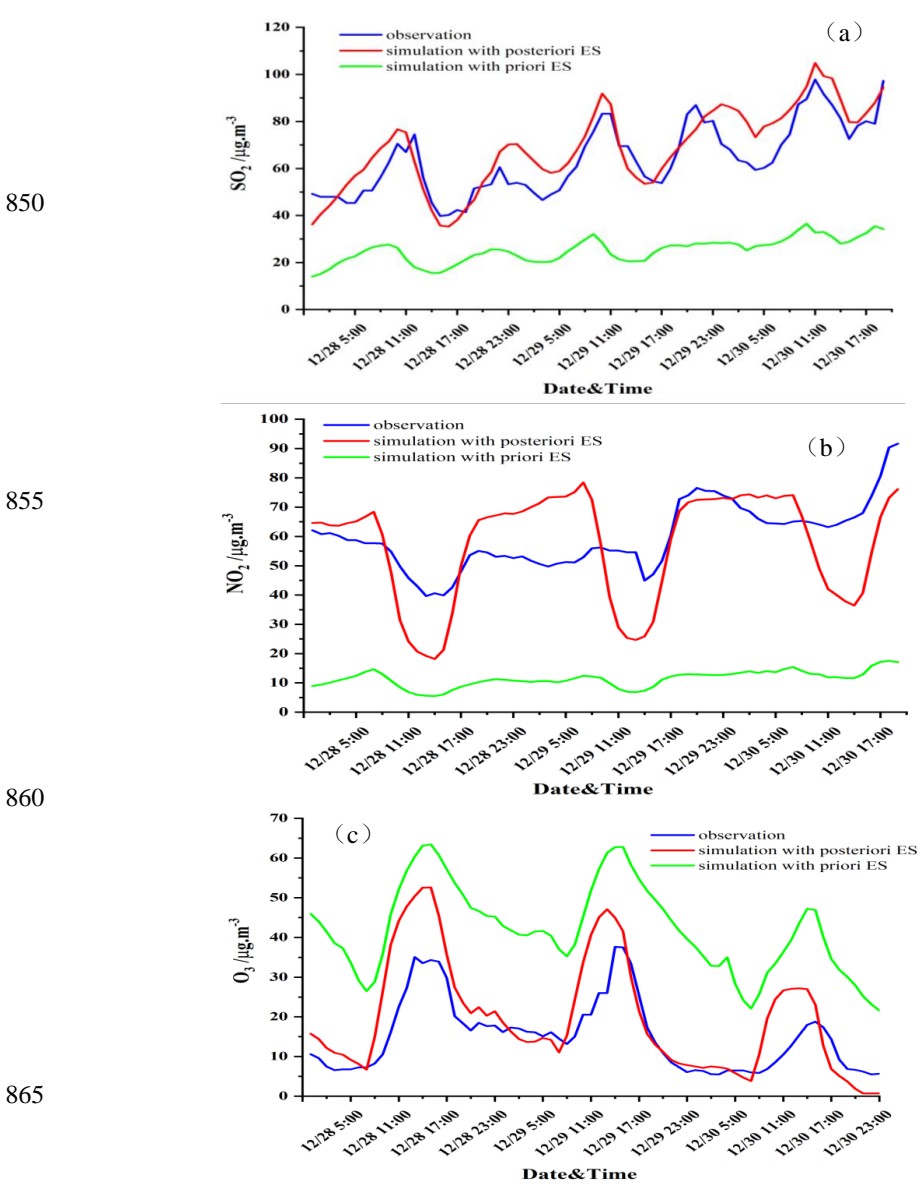

**Fig.11** Time serial of regional averaged (a) SO₂, (b) NO₂, and (c) O₃ concentrations respectively simulated with the *priori* and *posteriori* ES, and measurements at 45 stations in the BTH region during December 28–30, 2016.

870