# Peer review of "A New Inverse Modeling Approach for Emission Sources based on the DDM-3D and 3DVAR techniques: an application to air quality forecasts in the Beijing– Tianjin–Hebei Region"

_Atmospheric Chemistry and Physics, 2021_

## Author Comment (AC1)

**Response to Interactive comments from Anonymous Referee #1**

Referee comments are in black. Author responses are in blue.

General comments:

This study developed a new inverse modeling approach based on DDM-3D and 3DVAR, which was applied to assimilate surface SO2 and NO2 concentration measurements to optimize SO2 and NOx emissions in the Beijing-Tianjin-Heibei region. The a posteriori emission inventory is applied to air quality forecast, and results show the simulated surface SO2, NO2, and O3 concentrations using the a posteriori emission inventory are in better agreement with observations than a priori. I will recommend its publication after the following comments are addressed.

We thank the anonymous referee for his/her insightful and constructive comments. Below are our point-to-point responses in detail.

Specific comments:

1. Inversion modeling tools are summarized in the 3ʳᵈ paragraph in the introduction, but how to classify them is open to discussion. Is it appropriate to consider adjoint modeling and sensitivity analysis as two inversion modeling tools? In the manuscript Stavrakou et al. (2009) and Zhai et al. (2018) are considered as adjoint modeling. The two references did use adjoint modeling, but actually (1) Stavrakou et al. (2009) used adjoint modeling to calculate sensitivity, which was future used to minimize a 4DVAR cost function; (2) Zhai et al. (2018) only used adjoint modeling to calculate sensitivity without constrain emissions. Thus, Stavrakou et al. (2009) should be considered as 4DVAR, and Zhai et al. (2018) should not be cited. I agree that Henze et al. (2009) and Jiang et al. (2011) should be classified as 4DVAR; the two papers used GEOS-Chem adjoint model to calculate the sensitivity of cost function with respect to emissions scale factors, and the sensitivity were used to minimize cost functions. Considering sensitivity analysis as the name of inversion modelling is inappropriate as many approaches (such as 4DVAR) need to calculate sensitivity. Moreover, Mijling et al. (2012) was considered as sensitivity analysis in the manuscript, but actually it should be Kalman filter. I did not read all the references in the paragraph and could not help classify them; I suggest reading them carefully and classifying them properly.

   Many thanks for suggestions. It's right that considering the sensitivity analysis as one type of inversion modelling is inappropriate, we revise improper references and delete the type of sensitivity analysis according to suggestions.

2. Line 86: The reason that GEOS-Chem is not used urban air quality forecast is its spatial resolution is too coarse, and I suggest add this explanation.

   Corrected.

3. Line 88-90: Why it is stated EnKF and 4DVAR are absent of sensitivity analysis of the source-receptor relationship? Adjoint model (for example, GEOS-Chem adjoint) is used to calculate the source-receptor relationship in 4DVAR.

   Thanks for the suggestion. This sentence is inappropriate and we delete it.

4. Line 115-118: Are there any references to show that multiple receptors will result in high computational costs for adjoint model? My experience is whether assimilating one species or multiple species, the computational time difference for calculating the cost functions with respect to emissions is very small.

   Thanks for the comment. Differences of computational costs between multiple receptors and multiple species are lager, because sensitivity of the source-receptor relationship at all grids is calculated one receptor by one receptor, while sensitivity coefficients at all grids will be computed simultaneously for one species or multiply species. We add the related references in the revised manuscript.

5. How the uncertainties in Table 1 are calculated? If they come from other paper, please add references.

   We calculate the uncertainties based on activities and emission factors from the references using the Monte-Carlo method. We add the related descriptions and references in the revised version.

6. I suggest add some details of random perturbation method that is used to generate the 30 sets of inventories.

   Thanks for the suggestion. Firstly we calculate the uncertainties of $SO_2$ and $NOx$ emission sources, and we obtain the probability distribution of uncertainties for four sections of emission sources, respectively. Then we conduct thirty times of random perturbation on uncertainties of four sections of emission sources according to the probability distributions using the same perturbation coefficients for every perturbation. Lastly we calculate the total emission rates using random uncertainties of four sections for 30 sets of inventories, respectively. We add the detailed descriptions and the reference about random perturbation method in the revised manuscript.

7. Line 193: What "24-h strengths of ES for each month" mean?

It means that 24 hours diurnal variation of ES for every month. We revised it.

8.  How a priori SO2 and NOx emissions are vertically distributed in the model?

    We set 32 vertical levels in the CMAQ model and half of them is lower than 2km and other levels are located from 2km height to the top of atmosphere.

9.  Please check Eq. (3) careful. Which p should or not be capitalized?

    Thanks. P and ε are all revised to small letter.

Technical corrections:

1.  Line 74: Henze et al., 2008 -> Henze et al., 2009. And the year is also wrong in the reference list.

    Thanks. Revised.

2.  Please check Eq. (3) and corresponding descriptions carefully. Which p should or not be capitalized?

    Thanks. P and ε are all revised to small letter. The corresponding descriptions are corrected.

[revised manuscript text omitted]

---

## Author Comment (AC2)

**Response to Interactive comments from Anonymous Referee #2**

Referee comments are in black. Author responses are in blue.

Comments to "A New Inverse Modeling Approach for Emission Sources based on the DDM-3D and 3DVAR techniques: an application to air quality forecasts in the Beijing-Tianjin-Hebei Region"

General comments:

Timely precise emissions of air pollutants are crucial for air quality prediction and mitigation. The authors present a newly developed emission inversion method based on the combination three-dimensional decoupled direct (DDM-3D) and 3DVAR data assimilation techniques. The emission inversion method is applied to update the SO2 and NOx emissions over the Beijing-Tianjin-Hebei region during a heavy haze period. Their results demonstrate the newly updated emissions are reasonable and helpful to the prediction of the air pollutants including O3. The manuscript is well-organized and scientifically sound. Therefore, I recommend accepting it after minor revision.

We thank the anonymous referee for his/her insightful and constructive comments. Below are our point-to-point responses in detail.

Specific comments:

L180 Please describe the random perturbation method more detail.

Thanks for the suggestion. Firstly we calculate the uncertainties of $SO_2$ and NOx emission sources, and we obtain the probability distribution of uncertainties for four sections of emission sources, respectively. Then we conduct thirty times of random perturbation on uncertainties of four sections of emission sources according to the probability distributions using the same perturbation coefficients for every perturbation. Lastly we calculate the total emission rates using random uncertainties of four sections for 30 sets of inventories, respectively. We add the detailed descriptions and the reference about random perturbation method in the revised manuscript.

L203 Matrix **D** can not represent the impacts of local emissions at one grid on other grids. It should be **C**.

Thanks. It's mistake. Revised.

L270 Please change the Ls here and in formula (5), since you also use Ls in Line 214.

It's the same variable to Ls in line 214.

Figure 6 What are the reasons for the large discrepancies of the simulation and sensitivity coefficient over December 29?

Thanks for the comment. Large discrepancies of the simulation and sensitivity coefficient over December 29 may be related with absent calculation of high-order sensitivity coefficient in this case. In the future, we will adopt high-order sensitivity coefficient to improve constraint effects of $SO_2$ and $NO_x$ emission sources with the newly developed method in this study. We add the related descriptions in the discussion part of the revised manuscript.

[revised manuscript text omitted]